# Dual Targeting of Cancer Cells and MMPs with Self-Assembly Hybrid Nanoparticles for Combination Therapy in Combating Cancer

**DOI:** 10.3390/pharmaceutics13121990

**Published:** 2021-11-23

**Authors:** Kai Zhang, Jingjing Li, Xiaofei Xin, Xiaoqing Du, Di Zhao, Chao Qin, Xiaopeng Han, Meirong Huo, Lei Yang, Lifang Yin

**Affiliations:** Department of Pharmaceutics, School of Pharmacy, China Pharmaceutical University, Nanjing 210009, China; kaizhang@stu.cpu.edu.cn (K.Z.); lijingjing_9156@stu.cpu.edu.cn (J.L.); xxin@cpu.edu.cn (X.X.); duxiaoqing@stu.cpu.edu.cn (X.D.); dzhao_cpu@stu.cpu.edu.cn (D.Z.); nada77@cpu.edu.cn (C.Q.); xiaoph@cpu.edu.cn (X.H.); huomeirongcpu@cpu.edu.cn (M.H.)

**Keywords:** hybrid nanoparticles, transferrin, matrix metalloproteinases, immunomodulator, chemoimmunotherapy

## Abstract

The co-delivery of chemotherapeutic agents and immune modulators to their targets remains to be a great challenge for nanocarriers. Here, we developed a hybrid thermosensitive nanoparticle (TMNP) which could co-deliver paclitaxel-loaded transferrin (PTX@TF) and marimastat-loaded thermosensitive liposomes (MMST/LTSLs) for the dual targeting of cancer cells and the microenvironment. TMNPs could rapidly release the two payloads triggered by the hyperthermia treatment at the site of tumor. The released PTX@TF entered cancer cells via transferrin-receptor-mediated endocytosis and inhibited the survival of tumor cells. MMST was intelligently employed as an immunomodulator to improve immunotherapy by inhibiting matrix metalloproteinases to reduce chemokine degradation and recruit T cells. The TMNPs promoted the tumor infiltration of CD3+ T cells by 2-fold, including memory/effector CD8+ T cells (4.2-fold) and CD4+ (1.7-fold), but not regulatory T cells. Our in vivo anti-tumor experiment suggested that TMNPs possessed the highest tumor growth inhibitory rate (80.86%) compared with the control group. We demonstrated that the nanoplatform could effectively inhibit the growth of tumors and enhance T cell recruitment through the co-delivery of paclitaxel and marimastat, which could be a promising strategy for the combination of chemotherapy and immunotherapy for cancer treatment.

## 1. Introduction

The role of immune system is highly regarded in oncotherapy [1,2]. The complex interactions between tumor cells and the immune system run throughout the occurrence, progression and the regression of the cancer, which makes it possible to cure cancer through immunotherapy [3]. In recent years, several drugs for immunotherapy approved by FDA have obtained exciting clinical results in cancer treatment, such as anti-PD-1/PD-L1 monoclonal antibodies and CAR-T therapy [4,5,6]. Notably, most of the immune therapies are focused on T cells, which play the key role in the anti-tumor immunological effect [7]. T cells have the robust ability to directly kill tumor cells, and can influence tumor progression by cytokine secretion and cell interaction [8,9]. However, in the field of solid tumors, such as breast cancer, a leading cause of cancer-related deaths in women worldwide, immunotherapies including therapy focused on T cells still face great challenges. Breast cancer was considered as having a poor immunogenicity malignancy and has not been widely researched due to its insensitivity to immunotherapy [10]. The problems of high tumor load, cell recognition and cell chemotaxis make it hard for immunotherapy alone to obtain a satisfactory curative effect [3,11]. Chemotherapeutic agents not only kill the tumor cells and alleviate the tumor burden itself, but also release tumor antigens which help T cells recognize tumor cells [12,13]. Other benefits include regulating T cell function, reducing T_reg_ cells and modeling the tumor environment [14,15,16]. Taken together, there is a strong and developing case for combining chemotherapy and immunotherapy in breast cancer treatment [17]. Much scientific research and many clinical trials have proven the potency and advantages of the combination of chemotherapy and immunotherapy [18,19]. However, the problems of dosage regimen and delivery method bring new challenges.

Matrix metalloproteinase (MMPs) are the main extracellular proteinases responsible for degrading collagen and other proteins in the extracellular matrix (ECM) [20,21]. They have always been found to be overexpressed in tumor tissue, but not healthy tissue [22,23,24]. Plenty of evidence has demonstrated that MMPs not only regulate the ECM deposition in tumor microenvironment (TME), but also play a vital role in immune regulation and tumor immune escape [25,26]. For example, MMP-9 cleave or degrade chemokines such as CXCL9, CXCL10 and CXCL11. These chemokines can recruit CXCR3-positive T cells to tumor sites, which is an important step of anti-tumor immunity [27,28,29,30]. A further study even reported that MMPs can cleave almost all human chemokines [31]. Therefore, many researchers believe that modulating MMP levels in the tumor microenvironment can improve the immune response to tumor [32,33]. Marimastat (MMST) is the first broad-spectrum synthetic matrix metalloproteinase inhibitor [34]. It can mimic the substrates of MMPs to inhibit their activity reversibly and potently, even at a nanomolar concentration [35,36]. In this study, MMST was proposed to be an immunoregulator and functions by inhibiting MMPs and reducing cytokine degradation to recruit T cells for cancer treatment.

Paclitaxel (PTX) is a first-line drug for clinical treatment of diverse solid tumors and has been proven to be an effective cytotoxic drug for breast cancer [37,38]. The synergistic effects of PTX with immunotherapy have also been proven by previous studies [39,40,41]. PTX can induce the apoptosis of tumor cells, leading to the release of large amounts of tumor antigens which can be presented by dendritic cells to help activate cytotoxic T lymphocytes (CTLs) [42]. It also increases the sensitivity of immunosuppressive T_reg_ cells to PTX-induced apoptosis and reduces their numbers [43,44]. Other immune cells such as dendritic cells, natural killer cells (NK cells) and macrophages can benefit from PTX and enhance their anti-tumor capability as well [45].

Therefore, we decided to combine the treatment of MMST and PTX, expecting to achieve a synergistic effect of chemotherapy and immunotherapy to inhibit tumor growth in breast cancer. However, the precision delivery of immunoregulator and chemotherapeutic agents to the target site is the main challenge for the design of drug carriers [46,47]. To endow the PTX tumor cell selectivity, transferrin was used to load PTX and form a nanocomplex (PTX@TF) which targeted the transferrin receptor (TFR) overexpressed on tumor cells [48,49,50]. The lysolipid-containing thermosensitive liposomes (LTSLs) were chosen as the vehicle for co-delivery of PTX@TF and MMST. This temperature-sensitive liposome is a kind of intelligent nano-carrier that can be stabled in peripheral circulation but undergoes phase transitions after hyperthermia treatment (HT) at the tumor site, which results in its decomposition and facilitates the rapid localized release of encapsulated drugs [51,52]. PTX@TF assembled on the surface of MMST-loaded LTSTs (MMST/LTSLs) spontaneously through hydrophobic interaction and formed hybrid nanoparticles (transferrin-mediated nanoparticles, TMNPs), which were designed to co-deliver MMST and PTX, respectively, to the TME and cancer cells for chemoimmunotherapy in breast cancer.

The preparation and theoretical mechanism of TMNPs for tumor treatment are illustrated in Figure 1. Benefiting from the special thermosensitive ability of the LTSLs, the TMNPs could accumulate and disintegrate in tumors triggered by mild hyperthermia at lesion, releasing the protein-based complexes PTX@TF and MMST in the liposomes immediately. Through inhibiting MMPs such as MMP-9, MMST can recruit CXCR3-positive T cells to focus on tumor due to the increased chemokines in the TME, and then exhibit a synergistic effect with chemotherapeutics such as PTX to achieve chemoimmunotherapy. MMST was ingeniously employed as an immunomodulator, which may prove to be a promising application field for MMP inhibitors. In summary, TMNPs can deliver chemotherapeutic drugs and immunomodulators to target cancer cells and TME, respectively, to achieve precise and effective combination therapy.

## 2. Materials and Methods

### 2.1. Materials

PTX with more than 98% purity was purchased from Yew Biotechnology Co. Ltd. (Wuxi, Jiangsu, China). Human transferrin (TF) with more than 98% purity was purchased from Sigma-Aldrich Co., Ltd. (Jiangsu, China). MMST was purchased from Nanjing Adooq Co., Ltd. (Nanjing, Jiangsu, China). 3-(4,5-dimethylthiazol-2-yl)-2,5-diphenyltetrazolium bromide (MTT), fluorescein isothiocyanate (FITC), and rhodamine B (RhoB) were obtained from Sigma-Aldrich Co. Ltd. (St. Louis, MO, USA). 1,1-dioctadecyl-3,3,3,3-tetramethylindotricarbocyaineiodide (DiR) was purchased from Aibixin (Shanghai, China) Biotechnology Co., Ltd. (Shanghai, China). 1,2-Dipalmitoyl-DL-α-phosphatidylcholine (DPPC), N-(Carbonyl-methoxypolyethylene glycol 2000)-1,2-distearoyl-sn-glycerol-3-phosphoethanolamine, sodium salt (DSPE-mPEG2000), dialysis tube (3500 Da), ultrafiltration device (3500 Da) and soybean phosphatidylcholine (S100PC) were purchased from Shanghai AVT Pharmaceutical Technology Co., Ltd. (Shanghai, China). IL-2, TNF-α and TGF-β mouse monoclonal antibody was purchased from MULTI science biotechnology Co., ltd. (Hangzhou, China). Phospho-STAT5 mouse monoclonal antibody was purchased from CST (Shanghai) Biological Reagents Co., Ltd. (Shanghai, China). Alanine aminotransferase Assay Kit and Aspartate aminotransferase Assay Kit was purchased from Nanjing Jiancheng Bioengineering Institute Co., Ltd. (Nanjing, China). CXCL-10 and MMP-9 mouse monoclonal antibody were purchased from Abcam (Shanghai) Trading Co., Ltd. (Shanghai, China). 4T1, fetal bovine serum, RPMI-1640, DMEM, trypsin, and penicillin-streptomycin solution were purchased from Nanjing KeyGEN Biotech Co., Ltd. (Nanjing, China). The antibody for T cell sorting was purchased from BioLegend, Inc. (San Diego, CA, USA). DAPI and Annexin V-FITC/PI staining kit were obtained from the Beyotime Institute of Biotechnology (Haimen, China). 

### 2.2. Cells and Animals

#### 2.2.1. Cell Culture

4T1 cells were cultured in RPMI 1640 medium containing a combination of 10% fetal bovine serum and 1% penicillin streptomycin, and cultured at 37 °C with 5% CO_2_ and 100% humidity. Cells were split by 0.25% trypsin-EDTA solution when they reached 70–80% confluence.

#### 2.2.2. Laboratory Animal Care

All the experimental animals (BALB/C mice, female, 18–20 g) were purchased from the Beijing Vital River Laboratory Animal Technology Co., Ltd. (Beijing, China) and were cared for in accordance with the Principles of Experimental Animal Care and Guide for the Care and Use of Laboratory Animals. All animal experiments were approved by the Institutional Animal Care and Use Committee of China Pharmaceutical University on March 22, 2021 (Code: 2021-11-007).

### 2.3. Preparation and Characterization of TMNPs

#### 2.3.1. Preparation of PTX@TF

PTX@TF were prepared by the solvent evaporation technique. In brief, an aqueous solution of TF (2 mg/mL) was incubated with PTX (pre-dissolved in ethanol) at a TF/drug molar ratio of 5:1. The mixture was treated by miniature ultrasonic probe at 200 W for 10 min, and then the complexes were centrifuged at 8000 rpm for 10 min to remove unloaded drug precipitates and ethanol through ultrafiltration (MWCO = 3500 Da). Finally, the solid PTX@TF were obtained after the lyophilization of the protein suspension in a freezing dryer (FD-2D, Boyikang, China) without any cryoprotectants or additives for 36 h. The fluorescently labeled FITC-PTX@TF and RhoB-PTX@TF were prepared via the same method. The freeze-dried PTX@TF was used in all subsequent experiments and redissolved with PBS (in vivo and stability experiments) and distilled water (other experiments) for further experiments.

#### 2.3.2. Preparation of LTSLs and Other Liposomes

LTSLs were prepared by a film hydration method according to our previous studies. The phospholipids were weighed according to a mass ratio of DPPC/1-STEPC/DSPE-mPEG2000 = 8.6:1:0.4 and dissolved in the organic phase. After removing the organic phase via a rotary evaporator, PBS containing a certain amount of MMST with a pH of 6.5 was used for hydrating. The liposome suspension was treated with an ultrasonic cell crushing apparatus and extruded by the filter membrane with pore size of 0.22 μm. The unloaded drug was removed through ultrafiltration (MWCO = 3500 Da) by centrifuging at 8000 rpm for 10 min. RhoB- or DiR-loaded LTSLs (RhoB, DiR-LTSLs) were prepared in the same way.

#### 2.3.3. Preparation of TMNPs

TMNPs were prepared by dissolving the freeze-dried PTX@TF in the MMST/LTSLs solution. The mass ratio of TF/lipid was 1:3. FITC-PTX@TF/RhoB-LTSLs NPs, PTX@TF/DiR-LTSLs NPs, RhoB-PTX@TF/LTSLs NPs and PTX@TF/RhoB-LTSLs NPs were prepared in the same way.

#### 2.3.4. Characterization of TMNPs

Size and PDI of TMNPs and other formulations were measured with a dynamic laser scatter instrument (Brookhaven Instruments, Holtsville, NY, USA) at room temperature based on the operating guidelines of DLS. The excess samples and liquid were removed and then dried. The morphologies of the samples were observed via transmission electron microscopy (TEM) examination under JEM -1230 TEM (Tokyo, Japan) with an accelerating voltage of 200 kV.

The drug loading efficiency was detected using a U-3000 HPLC system (Themo, Waltham, MA, USA). The conditions for determination of PTX by high-performance liquid chromatography (HPLC) have been described in previous reports [53]. 

#### 2.3.5. Fluorescence Resonance Energy Transfer (FRET)

In the FRET study, FITC and RhoB were used as donors and recipients, respectively. FITC-PTX@TF was fixed at 1 mg/mL and RhoB/LTSLs was added in different FITC/RhoB ratios, and the FITC/RhoB ratios were 1:1, 1:2 and 1:3, respectively. The emission spectra of these samples were recorded by a fluorescence spectrometer (RF-5301PC Shimazu, Japan) at room temperature with an excitation wavelength of 450 nm.

To observe the serum stability of TMNPs, 1 mL FITC-PTX@TF/RhoB-LTSLs were mixed with 4 mL 1640 medium containing 10% fetal bovine serum and incubated in shaker (SHA-C, Jintan, China) at 100 RPM at 37 °C. FRET ratio was measured using the following formula at various times by a multifunctional enzyme label meter (Polarstar Omega, Germany).
FRET Ratio = I_R_/(I_R_ + I_F_) × 100%

In which I_R_ is the acceptor emission and I_F_ is the donor emission.

#### 2.3.6. Drug Release of TMNPs

HPLC was used to study the thermosensitive release of TMNPs. 1.0 mL of MMST/LTSLs (containing 200 μg/mL MMST) or TMNPs was moved into a dialysis tube (MWCO 3500 Da) and stirred with 200 mL of PBS solution (pH 7.4) in a dissolution tester (RC806D, Tiandatianfa, China). The dissolution temperature was 37 °C or 42 °C and the stirring rate was 100 RPM. The release of MMST at different times was measured by HPLC method. Similarly, HPLC was used to study the drug release of PTX in TMNPs. 2.0 mL of PTX@TF or TMNPs (containing 100 μg/mL PTX) was transferred into a dialysis tube (MWCO 3500 Da) and stirred with 30 mL release media in a constant temperature water bath shaker (SHA-B, Jingxianglong, Changzhou, Jiangsu, China). 0.8 M sodium salicylate solution (pH 7.4) was used as release media [54]. The dissolution temperature was 37 °C or 42 °C and the stirring rate was 100 RPM. The release of PTX at different times were measured by HPLC method. 

### 2.4. TFR-Mediated Endocytosis and In Vitro Cytotoxicity

#### 2.4.1. Transferrin-Receptor-Mediated Targeted Delivery of PTX@TF

4T1 cells (1 × 10^5^ cells per well) were inoculated in 12-well plates and treated with PBS or 10 mg TF in advance. After pretreating cells for 1 h, pre-heated RhoB-PTX@TF/LTSLs with different RhoB concentrations was added into the 12-well plates. The cells were incubated at 37 °C for 4 h, washed with cold PBS three times and resuspended in 500 μL of PBS for flow cytometry analysis (Accuri C6, BD, New York, NYC, America). To confirm the results of flow cytometry, we performed CLSM observations. 4T1 cells (1 × 10^5^ cells per well) were inoculated in a glass dish for 48 h and precultured for 1 h with PBS or 10 mg TF. Pre-heated RhoB-PTX@TF/LTSLs were incubated in a serum-free 1640 medium with 0.5 μg/mL RhoB for 4 h at 37 °C. The cells were washed three times with cold PBS to remove the excess formulations, and examined by CLSM (LSM800, Carl Zeiss, Germany) after fixed at 4% paraformaldehyde for 20 min and stained with DAPI for 10 min. 

#### 2.4.2. In Vitro Cytotoxicity and Apoptosis

4T1 cells (1 × 10^5^ cells per well) were seeded in 96-well plates and cultured at 37 °C for 24 h. The cells were incubated with drug-loaded formulations and separate free drugs at different concentrations for 48 h. 20 μL of MTT (5 mg/mL) was incubated into the 96-well plates and co-incubated for 4 h. The upper medium was carefully taken away. Subsequently, the formazan which had deposited at the bottom of the wells was redissolved with 150 μL DMSO and the absorbance of each well was measured by a microplate reader (Multiskan FC, Thermo Fisher Scientific, Waltham, MA, USA) at 570 nm.

Cell apoptosis was detected by an Annexin V-FITC/PI-staining kit (Cat: C1062M; Beyotime). 4T1 cells (2 × 10^5^ cells per well) were seeded in a 6-well plate and cultured at 37 °C for 24 h. Then the cells were treated with different formulations for 48 h. The concentrations of PTX and MMST in all preparations were fixed at 5 μg/mL. The cells were washed three times with cold PBS to remove the excess formulations and culture medium. The treated cells were suspended in a 300 μL binding buffer and then stained with 10 μL Annexin V-FITC and 5 μL PI for 15 min in the dark. Flow cytometry was used to observe the cell apoptosis after staining.

### 2.5. In Vivo Tumor Targeting Ability and Biodistribution

The BALB/C mice were inoculated subcutaneously under the axilla with 1 × 10^6^ suspended 4T1 cells. The experiment began at 2 weeks after tumor cells were injected. 200 μL of PTX@TF/DiR-LTSLs or DiR-LTSLs or free DiR was injected into the mice at a fixed DiR dose of 0.5 mg/kg via the tail vein. At 0.5, 1, 2, 4, 8 and 24 h after injection, fluorescence images were obtained by an in vivo imaging system (IN-VIVO FX Pro, Carestream, Canada) after mice were anesthetized. Finally, the mice were put down for their major organs and tumors. Fluorescence intensity in various organs was systematically calculated by an in vivo imaging system.

### 2.6. In Vivo Anti-Tumor Activity

#### 2.6.1. T Cell Sorting

The tumor tissues were ground and filtered to obtain cell suspensions. The cells were collected by centrifugation at 1000 RPM for 5 min and washed with PBS three times, and the supernatant was removed by centrifugation. Then cells were fixed via a cell fixation solution for 30 min and permeabilized by 1% Triton-100 at 37 °C for 10 min, followed by antibody incubation (CD3-PE: 0.25 μg/Test; CD4-FITC: 0.25 μg/Test; CD8a-FITC: 1 μg/Test; CD25-Cy7: 0.25 μg/Test; Foxp3-APC: 1 μg/Test, BioLegend, San Diego, CA, USA) at 37 °C in the dark for 30 min. The cell phenotypes were detected by flow cytometry after 500 μL PBS was added.

#### 2.6.2. Enzyme Linked Immunosorbent Assay (ELISA)

IL-2, TNF-α and TGF-β levels in cell culture supernatant or tissues were measured using mouse IL-2, TNF-α and TGF-β ELISA Kits (MultiSciences, Mouse IL-2 Elisa Kit, Cat: 70-EK202/2-96; MultiSciences, Mouse TNF-α High Sensitivity Elisa Kit, Cat: 70-EK282HS-96; MultiSciences, Human/Mouse/Rat TGF-β1 Elisa kit, Cat: 70-EK981-96) according to the manufacturer’s instructions.

#### 2.6.3. Protein Expression

The protein expression was determined by Western blot. The tumor tissue was homogenized in cold RIPA lysate with protease inhibitor (PMSF, Sigma-Aldrich, Shanghai, China), then the homogenate was centrifuged at 15,000 g for 20 min at 4 °C. The supernatant was collected and added with loading buffer to prepare the samples. Then the protein samples were loaded in the wells of SDS-PAGE gel, underwent electrophoresis and transferred to PVDF membrane. The PVDF membrane was incubated with primary antibodies at 4 °C overnight and a secondary antibody for 4 h, and then rinsed with Tris Buffered Saline Tween 20 (TBST) five times. Finally, the expression of different proteins was captured by a gel imaging system (Tanon-5200, Tanon Biotech, Nanjing, Jiangsu, China).

#### 2.6.4. Immunohistochemistry and Image Analysis

Dissected tumors were fixed in 4% (*w*/*v*) paraformaldehyde for 24 h and embedded in paraffin. Paraffin blocks were sectioned at 5 μm intervals using a paraffin microtome. For paraffin sections, slides were deparaffinized and rehydrated first, followed by destroying endogenous peroxidase activity with H_2_O_2_ (3 mL 30% H_2_O_2_ in 200 mL methanol) for 30 min at room temperature. Then, sections were boiled in a citrate acid buffer (10 mM, pH 6.0) via a heating mantle at 98 °C for 10 min for antigen retrieval. After cooling down, 50 μL of peroxidase blocking solution (Keygen, Nanjing, Jiangsu, China) was added to each section to block the activity of endogenous peroxidase, and the slices were incubated at room temperature for 10 min. Subsequently, 50 μL of non-immune animal serum (Keygen, China) was added to each section, and incubated at room temperature for 10 min. The serum was shaken and the sections were incubated with primary antibodies (BioLegend, San Diego, CA, USA) in a biotin blocking solution at 4 °C overnight in a humidified chamber. After this, the primary antibodies were aspirated and washed in PBS for 5 min at room temperature, five times. This was followed with 50 μL biotin-labeled second antibody (Keygen, Jiangsu, Nanjing, China) and Streptomyces antibiotin-peroxidase solution (Keygen, Jiangsu, Nanjing, China) incubated at room temperature for 10 min. Each section had 50 μL of freshly prepared DAB (1:200, Cat: PK6100; Vector Laboratories) added to it, observed under a microscope for 3–5 min, washed with ultra-pure water, and re-stained with 50 μL hematoxylin (Cat: GHS116; Sigma, Shanghai, China) for 15 s at room temperature. Finally, the sections were dehydrated with gradient alcohol and sealed with a neutral gum, and observed under an optical microscope (Ts2R, Nikon, Tokyo, Japan). All the integral optical densities (IOD) were quantified by Image Pro Plus (v 5.0) [55,56].

### 2.7. In Vivo Anti-Tumor Activity

#### 2.7.1. In Vivo Therapeutic Efficacy

Tumor-bearing mice were randomly assigned to each group after 10 days of being inoculated subcutaneously under the axilla with 1 × 10^6^ suspended 4T1 cells. Each group was treated with different formulations via tail vein injection at a dose of 5 mg/kg PTX and/or at 5 mg/kg of MMST seven times every 3 days. The MMST/LTSLs + HT and TMNPs + HT group were given hyperthermia treatment after intravenous administration by keeping the tumor site in a 42 °C water bath for one hour, while other parts of the body were thermally insulated. The tumor volume was calculated according to the below formula every 2 days, and the body weight was also measured every 2 days. At the end of the treatment, the mice were euthanized and the tumors were collected for subsequent experiments.
Tumor volume (mm^3^) = (Tumor Length (mm) × Tumor Width (mm)^2^)/2

#### 2.7.2. H&E and TUNEL Staining

The histomorphology and cell apoptosis of tumors was detected using H&E (Cat: C0105S; Beyotime, Nanjing, Jiangsu, China) and TUNEL kits (Cat: C1091; Beyotime, Nanjing, Jiangsu, China), respectively. The isolated tumor tissues were fixed in 4% paraformaldehyde and were paraffin-embedded to prepare 5 μm sections. Then, the H&E and TUNEL staining was performed according to the instructions. The sections were observed and photographed by optical microscope (Ts2R, Nikon, Tokyo, Japan) to provide a final report in the five representative fields. 

### 2.8. In Vivo Biocompatibility of TMNPs

H&E analysis was used to investigate the biocompatibility of TMNPs in vivo. Healthy BALB/c mice were randomly assigned to each group (n = 5). All the mice treated with different preparations (PTX@TF, MMST/LTSLs, TF/LTSLs, and TMNPs) were injected into tail vein (saline was used for the control group) seven times every 3 days. Finally, the mice were euthanized to isolate the major tissues for H&E analysis. The blood was sampled and centrifuged at 4000 RPM for 10 min to separate the serum. The levels of ALT and AST in the serum were measured using an Alanine aminotransferase Assay Kit (Nanjing Jiancheng, Cat: C009-2-1) and an Aspartate aminotransferase Assay Kit (Nanjing Jiancheng, Cat: C010-2-1) according to the manufacturer’s instructions.

### 2.9. Statistical Analysis

All data are shown as means ± standard deviations (SD). Statistical analysis was performed by one-way ANOVA. Significant differences between groups were set at * *p* < 0.05, ** *p* < 0.01, and *** *p* < 0.001 respectively. *p* < 0.05 was considered statistically significant in all analyses.

## 3. Results

### 3.1. Preparation and Characterization

PTX@TF complexes were prepared by a precipitation–ultrasonication method. The loading rate (%) of PTX in the complexes was 16.7% [57]. Film hydration was used to prepare the MMST/LTSLs and a supersonic probe was used to disperse the liposomes [58]. TMNPs were self-assembled by PTX@TF and MMST/LTSLs through electrostatic adsorption. The encapsulation efficiency of MMST in MMST/LTSLs was 56.72% analyzed by HPLC, with the ratio of drug to lipid being 1:10. Finally, the TMNPs were obtained by the self-assembly of PTX@TF with MMST/LTSLs at a TF/lipid mass ratio of 1:3. As shown in Table 1, the diameter of TMNPs was around 120 nm with a polydispersity index (PDI) of 0.23 and a slight negative surface charge of −1.62 mV. Due to the coating of PTX@TF, the average particle size and PDI of TMNPs were slight lager than those of MMST/LTSLs (size ≈ 90 nm, PDI ≈ 0.21) (Figure 2A). The TEM showed that MMST/LTSLs and TMNPs were symmetrical and spherical nanoparticles (Figure 2B).

In general, transferrin can be modified to the surface of cationic liposomes via crosslinking or electrostatic adsorption to obtain transferrin-receptor-mediated tumor targeting ability [59,60]. In this paper, the PTX@TF was self-assembled on liposomes via electrostatic adsorption to facilitate the dissociation of PTX@TF from LTSLs. To validate the self-assembly effect and stability of PTX@TF and MMST/LTSLs [61,62], a measurement of FRET was performed to determine whether PTX@TF could be self-assembled with MMST/LTSLs.

FITC was chosen as a fluorescence donor and then combined with PTX@TF, and RhoB was chosen as a fluorescence recipient and then encapsulated into LTSLs to prepare the FITC-PTX@TF/RhoB-LTSLs nanoparticles. The fluorescence intensity of FITC-PTX@TF/RhoB-LTSLs with various ratios were displayed in Figure 2C. The fluorescence intensity of the recipient RhoB was increased, whereas the fluorescence of donor FITC declined when the concentration of the RhoB/LTSLs increased, indicating that the donor produced a profound fluorescence response energy transfer to the recipient. On the contrary, with the decomposition of LTSLs during mild hyperthermia, the FITC-PTX@TF and RhoB/LTSLs separated, leading to the downward trend of receptor fluorescence and the upward trend of donor fluorescence (Figure 2D). In order to verify whether the link between the PTX@TF and MMST/LTSLs could remain stable in the peripheral blood, the binding stability of PTX@TF and MMST-LTSL was also studied by FRET. After being stored in a cell culture medium with 10% FBS, the FRET ratio of FITC-PTX@TF/RhoB/LTSLs showed no obvious change within 24 h (Figure 2E), which indicated that TMNPs were stable in the blood circulation system. In summary, the measurements of FRET further demonstrated that the PTX@TF could be assembled on LTSLs and was stable for at least 24 h.

We hypothesized that TMNPs remained intact until they reach the tumor site after intravenous injection. Further, after being stimulated by local hyperthermia treatment, TMNPs released the encapsulated MMST into the tumor microenvironment to modulate the immunosuppressive state due to the thermosensitivity of LTSLs. To investigate the thermosensitivity of MMST/LTSLs and TMNPs, drug release experiments at 37 °C and 42 °C were performed for both MMST/LTSLs and TMNPs by HPLC to determine the release profiles of MMST every 10 min. There was no difference between the MMST/LTSLs and TMNPs, indicating that PTX@TF assembled on LTSLs had a negligible effect on the release of MMST (Figure 2F). Nonetheless, due to the presence of lysolipids in the lipid bilayer of LTSLs, both LTSLs and TMNPs showed a faster release at 42 °C compared to that at normal body temperature of 37 °C. This demonstrated that the TMNPs and MMST/LTSLs had similar characteristics. Apart from that, TMNPs were capable to release-loading drugs and PTX@TF, which coated on the surface of a liposome after hyperthermia treatment at the tumor site. The stability of PTX@TF was critical for the cellular uptake of chemotherapy drugs. Therefore, in vitro drug release of PTX in PTX@TF and TMNPs was performed by HPLC. As shown in Appendix A, the drug release of PTX in PTX@TF and TMNPs at 42 °C or 37 °C had a similar release curve and release time caused for almost 36 h. Higher temperatures slightly accelerated the release of PTX. This result indicated that the immunomodulator MMST was well encapsulated in TMNPs after injection into the circulating system and was rapidly released at disease sites after HT treatment. At the same time, the nano-suspension PTX@TF could be stably packaged on the LTSLs in the circulating system and dissociated from LTSLs after HT treatment, and ensure the stability of cytotoxic drug loading. TMNPs were proved to be a sensitive and stable nanocarrier for the separate co-delivery of chemotherapeutic agents and immune modulators to their respective targets. The successful construction of TMNPs was crucial for subsequent experiments.

### 3.2. Transferrin Receptor Mediated Uptake of PTX@TF

After the TMNPs were disrupted by HT treatment and released via the payloads encapsulated in LTSLs, MMST and PTX@TF were dispersed into TME. PTX@TF could be quickly internalized by cancer cells mediated by transferrin receptors. The cellular uptake mediated by the interaction between transferrin and transferrin receptors is the key to exert the chemotherapeutic effect of PTX@TF. To verify this hypothesis, RhoB-PTX@TF was fabricated, incubated with 4T1 cells and monitored by a confocal laser scanning microscope (CLSM) and flow cytometry (Figure 3A,B). As time went on, the fluorescence intensity became stronger and the cellular uptake of RhoB-PTX@TF by 4T1 cells were enhanced. These results confirmed that PTX@TF was effectively taken up by 4T1 cells.

In order to study the role of transferrin receptors in cellular uptake, RhoB-PTX@TF were incubated with transferrin receptors saturated 4T1 cells by excessive native transferrin. After a 4 h incubation with different concentrations of TF, the cellular uptake of RhoB-PTX@TF was measured by flow cytometry. As showed in Figure 3C, the uptake of RhoB-PTX@TF by regular 4T1 cells was significantly higher than that of preincubated 4T1 cells with TF, when the TF concentration was higher than 1 mg/mL. Similarly, CLSM observation also revealed a much lower distribution of RhoB-PTX@TF (red fluorescence) in the pretreated cells than in the untreated cells (Figure 3D). These results suggested that the uptake of RhoB-PTX@TF was accomplished by TFR-mediated endocytosis in breast cancer cells.

### 3.3. In Vitro Anti-Tumor Effect

In contrast to free PTX, the survival rate of 4T1 cells in the PTX@TF group was similar to the free PTX group when the concentration of PTX ranged from 1 to 10 μg/mL, which indicated that the efficacy of the cytotoxic drugs was not affected by the interaction with transferrin (Figure 3E). Based on our previous research, we confirmed that the best anti-tumor effect was achieved when the mass ratio of PTX/MMST was 1:1 [58]. Therefore, TMNPs with a PTX/MMST ratio of 1:1 was used to confirm the toxicity in 4T1 cells. As expected, TMNPs showed greater toxicity than PTX@TF when the concentration of PTX was higher than 1 μg/mL. Additionally, the toxicity of TMNPs became more significant after HT treatment (Figure 3E), implying that the temperature sensitivity of TMNPs affects this as well. This result was confirmed by the apoptosis analysis with Annexin V-FITC/PI double staining monitored by flow cytometry (Figure 3F). Compared with the 4T1 cells treated with PBS, the cell apoptosis rate of those treated with TMNPs or TMNPs + HT was significantly enhanced, which indicated that TMNPs can effectively promote the apoptosis of tumor cells (Figure 3G). Interestingly, the cell apoptosis rate of TMNPs was significantly higher than PTX@TF. In order to investigate the mechanism of the high cell apoptosis induced by TMNPs, MTT assay was performed in 4T1 cells incubated with MMST, MMST/LTSLs and MMST/LTSLs + HT at different concentrations. As shown in Appendix A, free MMST did not show significant cytotoxicity in 4T1 cells even at high concentrations, whereas the MMST/LTSLs and MMST/LTSLs with HT treatment (MMST/LTSLs + HT) displayed significant cytotoxicity at high MATT concentrations. It was mainly due to this that LTSLs were able to effectively deliver MMST to the TME and increase the bind between MMST and membrane-type MMPs (MT-MMPs), which presented in the cell membrane and regulated apoptosis of cancer cells by cleaving pro-apoptotic-related ligands or receptors [63,64,65]. Therefore, unlike free MMST, MMST/LTSLs and MMST/LTSLs + HT induced higher cytotoxicity and apoptosis rate in breast cancer cells (Figure 3F). In addition, the result of combination index showed that when the total concentration of TMNPs or TMNPs + HT was higher than 0.2 μg/mL (CI < 1, at a concentration of 0.1 μg/mL PTX or MMST), indicating the combination of the PTX@TF and MMST/LTSLs, they exhibited an obvious synergistic effect and hypothermia treatment further boosted the efficacy of TMNPs (Appendix A). In conclusion, TMNPs and TMNPs + HT showed prominent in vitro anti-tumor effects. 

### 3.4. In Vivo Tumor Targeting Ability and Biodistribution

Transferrin receptors (TFR) have been reported to be overexpressed 100-fold more in tumor cells than in normal cells, as iron supplementation is desired to maintain the metabolism, proliferation and survival of tumors [66,67]. Elevated TFR is closely associated with a poor outcome for breast cancer patients, making it an attractive molecule for the targeted therapy of breast cancer [68,69,70]. This provided the rationale to enhance the tumor targeting ability of TMNPs with the modification of PTX@TF. For the study of the biodistribution of TMNPs, LTSLs laded with a fluorescent probe DiR (DiR-LTSLs) were prepared. The dynamic behavior of PTX@TF/DiR-LTSLs can represent TMNPs to some extent because of the similarities in structure. Compared to free DiR, mice treated with DiR-LTSLs or PTX@TF/DiR-LTSLs achieved remarkably increased fluorescence intensity accumulated in tumors (Figure 4A,B). PTX@TF with one-step facile preparation significantly enhanced the accumulation of nanoparticles at the site of tumors by 1.6-fold via TFR-mediated endocytosis, compared to DiR-LTSLs at 8 h. The main tissues and tumors were collected at 24 h after injection. Compared to the mice treated with free DiR or DiR-LTSLs, the ex vivo fluorescence signal of tumor was significantly enhanced, demonstrating the excellent transferrin-mediated tumor targeting of TMNPs (Figure 4C,D). It is also important to note that although the TMNPs can significantly increase the accumulation of the loading drug at the tumor site, large accumulations of the DiR were also shown in other normal tissues (Figure 4C,D). It is mainly due to the characteristics of liposomes that make DiR-LTSLs and PTX@TF/DiR-LTSLs phagocytosed by abundant macrophages in some organs [71,72]. In addition, transferrin receptors were distributed in many cells, such as activated lymphocytes and serum-induced fibroblasts, that needed iron for normal growth and development, but not in tumor cells [73]. This resulted in the high accumulation of PTX@TF/DiR-LTSLs in some tissues, such as spleen and liver tissue [74]. 

### 3.5. In Vivo Immune Regulation and T Cell Recruitment

MMPs play a role in cancer-mediated immune suppression by regulating the response of T lymphocytes against tumor cells [75]. MMP-9 can inactivate or antagonize the biological functions of tumor-suppressing cytokines and chemokines by proteolytic cleavage and thereby jeopardize the efficacy of chemoimmunotherapy [76,77]. MMP-9 has been reported to cleave and potentially degrade T-helper cell 1 (Th1)-type chemokines like CXC ligand CXCL10 and inhibit their anti-tumor outcome, including the recruitment of CD4^+^ T cells and CD8^+^ T cells, and their tumor-killing effect [27,29,78]. To explore the mechanism of MMP inhibition and immune enhancement therapy of TMNPs in vivo, Western blot was used to investigate the expression levels of MMP-9 (92 kDa) and CXCL10 (10 kDa) in tumor tissues collected after treatment by saline, PTX@TF, MMST/LTSLs + HT and TMNPs + HT seven times every three days, respectively, in the 4T1 tumor-bearing BALB/C mouse model (Figure 5A). MMST/LTSLs + HT and TMNPs + HT inhibited the expression of MMP-9 and induced the increase of CXCL10 (Figure 5B,C). To further study the effects of TMNPs + HT on T cells, we investigated the expression of p-STAT5 (90 kDa), an indicator of T cell activation and proliferation in tumor-infiltrating lymphocytes by Western blot (Figure 5D). Consistent with our hypothesis, TMNPs + HT treatment increased the expression of p-STAT5, implying the improved activation and proliferation of T cells. 

To study the effects of TMNPs + HT on the immune system, we extracted lymphocytes from tumor tissues. Compared with the saline group, TMNPs + HT treatment increased the proportion of CD3^+^ cells by 108.5% (Figure 5E), CD4^+^ helper T cells by 73.9% and CD8^+^ cytotoxic T cells by 325.8%, respectively (Figure 5F,G), among tumor-infiltrating lymphocytes, suggesting that the MMST/PTX combination treatment promoted the functional T cell-mediated anti-tumor response. Meanwhile, lower T_reg_ cell abundance (CD4^+^CD25^+^FoxP3^+^) was obtained after TMNPs + HT treatment, which was mainly due to the inhibition of MMST on T_reg_ cell differentiation by suppressing the expression level of TGF-β (Figure 5H and Figure 6C,D) [79,80]. It was also noteworthy that PTX@TF decreased the percentage of T_reg_ cells by impairing their viabilities and tumor-suppressing functions rather than CD3^+^CD4^+^ helper T cells and CD3^+^CD8^+^ cytotoxic T cells (Figure 5F–H), which could further enable PTX to synergize with MMST to achieve a potent chemoimmunotherapeutic effect in the breast cancer. 

### 3.6. Immune-Related Cytokines in Tumors

T cells exposed to breast cancer cells can produce cytokines like IL-2, IFN-γ and TGF-β to mediate immune responses [81]. IL-2 associated with T cell proliferation can be activated by IL-2Rα signaling in vivo and can contribute to cancer immunotherapy [82]. IFN-γ is a multifunctional immunoregulatory cytokine, mainly produced by CD4^+^ T cells, CD8^+^ T cells and NK cells, and plays a crucial role in both innate and adaptive immunity [83,84]. However, certain MMPs like MMP-2 and MMP-9 can cleave IL-2Rα on the T cell surface, weaken the IL-2 function and then decrease the expansion of natural killer (NK) cells and T lymphocytes. On the other hand, MMPs regulate the activation of TGF-β to promote the recruitment of Treg cells, and reduce the activity of NK cells and CD8^+^ T lymphocytes, thus leading to tumor progression [85]. To examine the alternation of cytokine levels after TMNP treatment in the 4T1 tumor-bearing mouse model, tumor tissues were homogenized in PBS (0.01 M, PH 7.4) buffer and then supernatants were collected for analysis of cytokine levels by ELISA kit. Compared to the saline group, the levels of IL-2 and IFN-γ were significantly increased by 5.4-fold and 3.9-fold after treatment with TMNPs + HT (Figure 6A,B), which provided the supporting proof for the recruitment of cytotoxic T cells by MMP inhibition. Meanwhile, the level of TGF-β was significantly downregulated by 3.1-fold (Figure 6C) with accordingly reduced numbers of Treg cells (Figure 5H). An immunohistochemistry (IHC) assessment of IL-2, IFN-γ and TGF-β expression in tumor tissues (Figure 6D–G) showed a consistent result with ELISA analysis. These findings confirmed that TMNPs + HT were capable to potentiate the immune response and inhibit immunosuppressive activity in breast cancer through the downregulation of MMPs and the regulation of cytokine levels.

### 3.7. In Vivo Anti-Tumor Efficacy

The failure of treatments for breast cancer is generally due to multidrug resistance, low immunogenicity and intense immunosuppressive environment [86]. As a result, it is insufficient to treat breast cancer by chemotherapy alone. Therefore, the combination of chemotherapy and immunotherapy is a reasonable method for breast cancer treatment. The anti-tumor effects of different formulations were evaluated by tumor volume and tumor weight in a xenograft 4T1 tumor-bearing BALB/C mouse model (Figure 7A,B). The tumor volume of the control group significantly increased on day 28 after tumor inoculation (Figure 7A). Meanwhile, the tumor growth rate of mice treated with other formulations was significantly lower than that of the control group. Remarkably, the drug delivered via the nanocomplex (PTX@TF) or liposomes (MMST/LTSLs) could inhibit tumor growth better than the free drug (PTX and MMST). Notably, the mice injected with TMNPs and locally heated at the tumor site achieved the best anti-tumor therapeutic effect. The weight measurement of isolated tumors collected at the end of the experiment reinforced the highest anti-tumor efficacy of TMNPs, with 2.45- and 1.82-fold less than MMST/LTSLs and PTX@TF, respectively (Figure 7C,D). Importantly, TMNPs with HT treatment (TMNPs + HT) showed an enhanced anti-tumor efficacy compared to TMNPs without HT treatment, with 1.7- and 1.69-fold less in tumor volume and tumor weight, respectively, demonstrating that TMNPs have a significant temperature sensitivity and can achieve a reinforced anti-tumor efficacy after HT treatment.

In order to further evaluate the anti-tumor effect of TMNPs in vivo, tumor tissue sections were investigated. As shown in histological sections (Figure 7E), the tumor cells were dense in normal tumor tissue, but the intact morphology of tumor tissue was destroyed after treatment with different formulations. Among them, the TMNPs + HT group had the strongest tumor cell necrosis and the most obvious morphological changes. Ki67 immunohistochemistry and TUNEL staining also showed that TMNPs could significantly increase the apoptosis of tumor cells in tumor tissue and inhibit the proliferation of tumor cells (Figure 7G,F). All of these results suggested that co-delivery of PTX@TF and MMST/LTSLs by TMNPs can achieve enhanced anti-tumor efficacy in breast cancer. 

### 3.8. In Vivo Biocompatibility

LTSLs have displayed their fantastic safety and biocompatibility in our previous study and we did not observe significant cytotoxicity induced by TF during our in vitro anti-tumor efficacy study (Figure 3E,F) [58]. Nevertheless, it is concerning that PTX has been associated with the elevation of glutamic–pyruvic transaminase (ALT) and glutamic–oxalacetic transaminase (AST) ranging from 7% to 26% in cancer patients [87]. With a large number of TMNPs accumulating in the liver due to phagocytosis of the reticuloendothelial system (Figure 4C) [88], it is necessary to explore the biocompatibility of TMNPs to grant a sophisticated understanding for the safety and possible adverse effects of this hybrid thermosensitive delivery system. Healthy BALB/c mice were administrated with MMST/LTSLs, PTX@TF, TF/LTSLs and TMNPs at a PTX dose of 5 mg/kg or at 5 mg/kg of MMST, seven times every three days. Compared with the control group, the injection of these formulations displayed no obvious organ damage in H&E staining (Figure 7H). The serum levels of glutamic–pyruvic transaminase (ALT) and glutamic–oxalacetic transaminase (AST) were measured to further explore whether there were potentially toxic effects caused by MMST/LTSLs and TMNPs on the liver [89]. Compared with the control group, the levels of ALT and AST for the mice treated with MMST/LTSLs were slightly increased, suggesting that the accumulation of MMST did not cause much damage to the liver. (Appendix A) In addition, the levels of ALT and AST for PTX@TF and TMNPs treatment were increased by 1.41-fold and 1.38-fold, respectively, compared to the control group, which was a less than 3-fold upper change limit according to the Guidelines for the management of drug-induced liver injury [90,91,92]. These results indicated that no acute hepatic necrosis was induced by this delivery system, and TMNPs with robust in vivo anti-tumor activity and good tolerance had no significant toxic effects on major organs in mice.

In conclusion, successful chemoimmunotherapy was achieved through the versatile nanoplatform self-assembled by the transferrin-based nanocomplex and the thermosensitive liposome, which could intelligently deliver the chemotherapeutic PTX and the immunomodulator MMST to their action sites, separately but at the same time. The accurate targeting ability of the nanoplatform was endowed by the collapse of the vector under local hyperthermia treatment, leading to the release and separation of the PTX@TF and MMST. PTX@TF killed tumor cells with higher efficiency due to the enhanced cellular uptake mediated by transferrin. MMST interacted with the MMPs in the TME to reduce the chemokine degradation, which improved the immune microenvironment by increasing T cells and the inflammatory factor with its anti-tumor effect to work synergistically with PTX. Due to the smart design and the synergistic effect of the two payloads, TMNPs could exert better therapeutic effect in breast carcinoma.

## 4. Discussion

Hyperthermia treatment, as an adjuvant therapeutic modality to room temperature, is an effective means to greatly increase nanoparticle concentration in tumors by improving local blood and interstitial fluid flow [93]. Local drug delivery of MMST with thermosensitive liposomes and hyperthermia to target the tumor microenvironment has shown to achieve high local drug concentrations with good therapeutic efficacy in our previous study. Nevertheless, with the dissemblance of the thermosensitive liposomes, encapsulated chemotherapy agents like PTX can be released followed by weakened cellular uptake efficiency and poor tumor penetration due to the hydrophobicity. To solve this problem, synthesis procedures were explored to conjugate PTX with hyaluronic acid (HA-PTX) to promote its endocytosis by cancer cells [58], but it was time-consuming and difficult to scale-up for industry purpose. Inspired by the concept of protein-based delivery named “Drug-delivering-drug”, as we previously investigated [94], dual-functional transferrin was selected as the protein carrier for PTX and the targeting ligand for transferrin receptors overexpressed on cancer cells, to achieve a decent drug loading of 16.7% and a fine self-assembly with LTSLs by electrostatic interaction in this study (Figure 2A). The protein-based thermosensitive nanoplatform was characterized by its profound accumulation and penetration in tumor sites, which was recognized by TFR-mediated endocytosis in breast cancer cells (Figure 3A,D) and the significant tumor accumulation of TMNPs in 4T1 tumor-bearing mouse model (Figure 4).

The development and progression of breast cancer is influenced by multiple elements in the tumor microenvironment. We proposed the chemoimmunotherapeutic strategy by synergizing PTX with the immunomodulator MMST to recruit CD4^+^ T helper cells and cytotoxic CD8^+^ T cells (Figure 5), and to restrict the escape of cancer cells from immunosurveillance, as immune cells are considered to be the most important factor throughout breast carcinogenesis [95]. For the activation of immune cells with MMP inhibition by MMST, CXCL-10 stimulated immune cells through polarization and activation of Type 1 T-helper cells, resulting in the enhanced paracrine secretion of IFN-γ and IL-2 to facilitate the anti-tumor efficacy of TMNPs + HT (Figure 5A, Figure 6 and Figure 7). With a higher number of tumor-infiltrating lymphocytes after the treatment of TMNPs + HT (Figure 5F,G), it is promising to have more favorable outcomes after TMNPs + HT chemoimmunotherapeutic treatment including prognosis amelioration and prolonged survival in breast cancer patients. However, the autocrine CXCL9, -10, -11/CXCR3 signaling axis in cancer cells has the potential to increase proliferation, angiogenesis, and metastasis of cancer cells [96], which indicates the balance between pro-inflammation and anti-inflammation in the tumor microenvironment should be considered during chemoimmunotherapy. Poor prognostic scenarios could occur if pro-inflammation dominates the tumor microenvironment and the vicious transformational epithelial cells escape from immune surveillance [97]. Based on this, the most striking result of our study was that a better balance between pro- and anti-inflammatory effects as the best anti-tumor effect by TMNPs + HT treatment in 4T1 tumor-bearing mouse model was obtained through the alternation of cytokine levels by MMP inhibition and CXCL-10 promotion, and no tumor progression induced by autocrine CXCL9, -10, -11/CXCR3 signaling in TMNPs or TMNPs + HT groups was observed (Figure 6). Therefore, the sufficient activation of an immunosuppressive tumor microenvironment probably benefited from the protein-based complexes thermosensitive delivery system via transferrin-receptor-mediated endocytosis, synergistic effect of PTX and MMST in chemoimmunotherapy, and the optimized dose and ratio of PTX and MMST.

## 5. Conclusions

This study established a novel nanoparticle based on a transferrin-modified chemotherapy nanocomplex and thermo-sensitive liposomes for breast cancer treatment. We believe this hybrid nanoplatform can be a promising drug delivery system for cancer treatment by modifying the transferrin and liposomes with other active compounds, and could easily enable industrial production to meet the criteria for clinical trials due to its simple preparation process. In addition, we uncovered that MMP inhibitors could achieve anti-tumor efficacy by exerting anti-immunosuppression effects, but not anti-metastasis and anti-angiogenesis effects. These findings will help us expand the role of MMP inhibitors in cancer treatment. However, the accumulation of TMNPs in other normal organs is associated with minor liver damage. How to further improve the targeting ability or reduce the side effects of the hybrid nanoparticles will be the next problem that needs to be solved. In addition, the mechanism of the combined treatment of PTX and MMST also needs to be further explored. In conclusion, we fabricated a new nanoplatform assembled by a transferrin-based nanocomplex and thermo-sensitive liposomes for dual- targeting drug delivery, which is a potent tumor therapy strategy with good prospects.

## Figures and Tables

**Figure 1 pharmaceutics-13-01990-f001:**
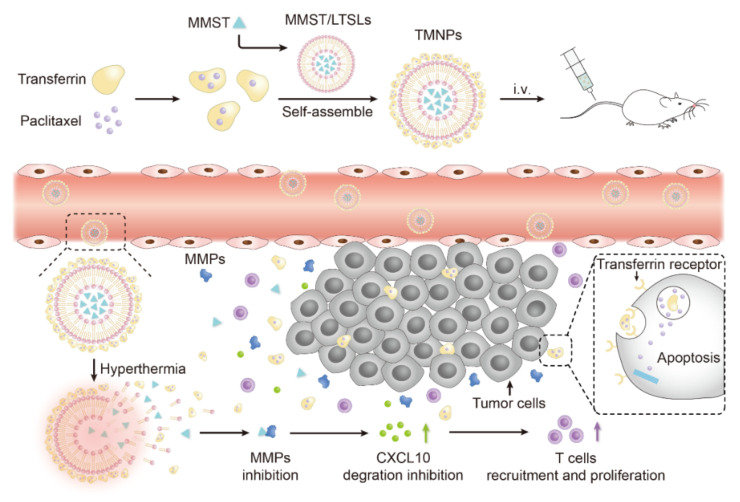
Schematic diagram of preparation of TMNPs and mechanism of chemoimmunotherapy. PTX@TF was prepared by precipitation-ultrasonication method and self-assembled on MMST/LTSLs surface to form hybrid nanoparticles (TMNPs). TMNPs accumulated in the tumor through EPR effect and transferrin-receptor-mediated endocytosis, and rapidly released the loaded MMST and PTX@TF by the stimulation of local hyperthermia treatment. The released MMST inhibited the activity of MMPs to reduce chemokine degradation and increased T cell proliferation and infiltration in the TME. Meanwhile, the nanocomplex PTX@TF entered tumor cells via transferrin targeting, leading to potent apoptosis of tumor cells with the facilitation of T cell recruitment.

**Figure 2 pharmaceutics-13-01990-f002:**
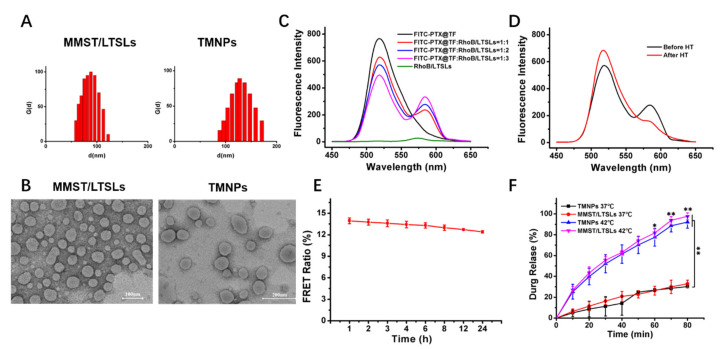
Characterization of TMNPs. (**A**) Size distributions of MMST/LTSLs and TMNPs. (**B**) TEM images of MMST/LTSLs and TMNPs (the scale bar was 100 nm and 200 nm, respectively). (**C**) Fluorescence spectra of fluorescence resonance transfer test with different ratios of FITC-PTX@TF/RhoB-LTSLs. (**D**) Fluorescence emission spectra of FITC-PTX@TF/RhoB-LTSLs with (red) and without (black) HT treatment at 42 °C. (**E**) Serum stability of FITC-PTX@TF/RhoB-LTSLs within 24 h (mean ± SEM, *n* = 3). (**F**) In vitro drug release of MMST in MMST/LTSLs and TMNPs at 42 °C or 37 °C. (mean ± SEM, *n* = 3, ** *p* < 0.01).

**Figure 3 pharmaceutics-13-01990-f003:**
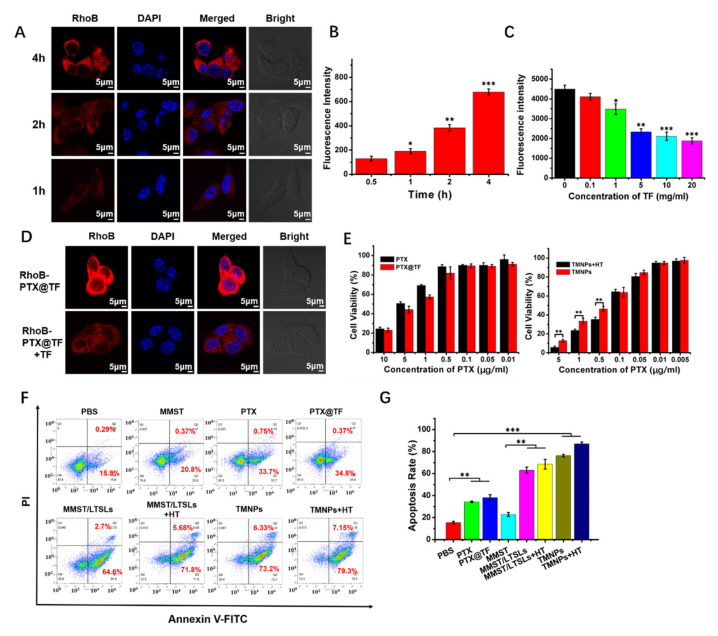
TFR-mediated endocytosis and in vitro cytotoxicity. (**A**) CLSM images and (**B**) fluorescence intensity of RhoB-PTX@TF in 4T1 cells after incubation for 0.5 h, 1 h, 2 h and 4 h at 37 °C. (**C**) Cellular uptake of RhoB-PTX@TF (red) with preincubation by different TF concentrations in 4T1 cells after 4 h incubations at 37 °C. The fluorescence intensity was measured by flow cytometry. (**D**) CLSM observation of cellular uptake. RhoB-PTX@TF (red) were treated with 4T1 cells with or without preincubation with 10 mg/mL TF for 4 h at 37 °C. The scale bar is 5 μm. (**E**) Cell viability after treatment with PTX, PTX@TF and TMNPs, and TMNPs by at 37 °C. PTX had an equal concentration to MMST in TMNPs. (**F**) Flow cytometry analysis and (**G**) quantitative analysis of apoptosis induced by PBS, PTX, MMST, PTX@TF, MMST/LTSLs, MMST/LTSLs with HT (MMST/LTSLs + HT), TMNPs, and TMNPs with HT (TMNPs+HT) for 48 h at 37 °C. All the formulations were prepared at a PTX and/or a MMST concentration of 5 μg/mL. Cells were stained with FITC-Annexin V and PI (mean ± SEM, *n* = 3, * *p* < 0.05, ** *p* < 0.01, *** *p* < 0.001).

**Figure 4 pharmaceutics-13-01990-f004:**
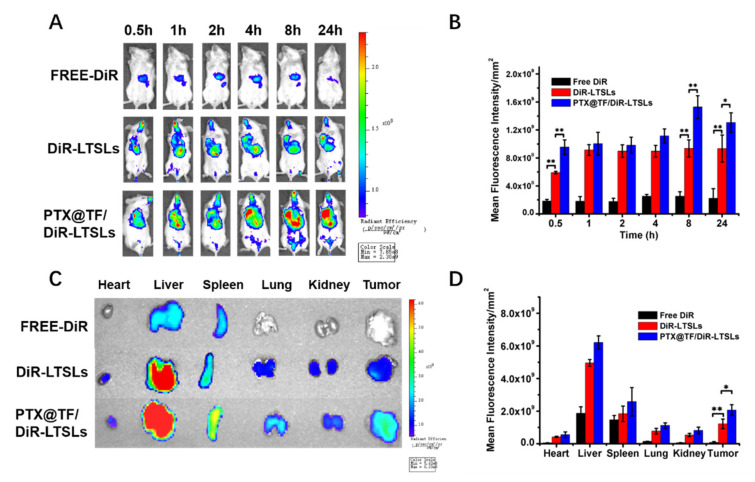
Tumor targeting ability of TMNPs in vivo. 4T1 tumor-bearing BALB/C mice were randomly assigned into three groups (three mice per group) and injected with free DiR, DiR/LTSLs or PTX@TF/DiR-LTSLs at a fixed concentration of 0.5 mg/kg DiR. The PTX@TF/DiR-LTSLs were used to replace the TMNPs for better observation. Free DiR was used as control. (**A**) The representative in vivo fluorescence imaging collected at 0.5, 1, 2, 4, 8, 24 h. (**B**) Quantification of fluorescence intensity (DiR) at the tumor site (n = 3, * *p* < 0.05, ** *p* < 0.01). (**C**) Ex vivo fluorescence images of tissues, including tumor, heart, liver, spleen, lung, and kidneys harvested at 24 h post-injection of different formulations. (**D**) Quantification of fluorescence intensity (DiR) in the major organs and tumors (*n* = 3, * *p* < 0.05, ** *p* < 0.01).

**Figure 5 pharmaceutics-13-01990-f005:**
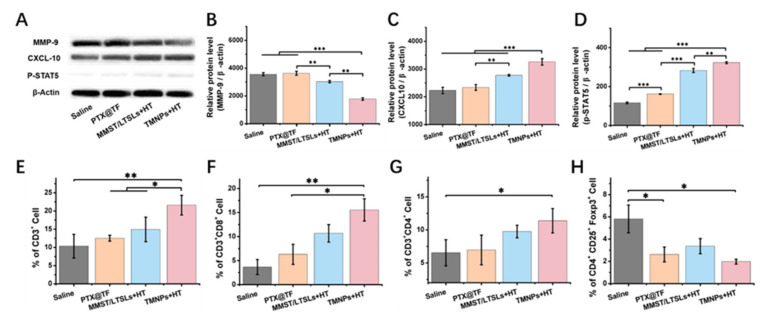
T cell recruiting ability of MMST. 4T1 tumor-bearing mice were euthanized to separate breast tumor at day 29 after saline, PTX@TF, MMST/LTSLs + HT and TMNPs + HT treatment, seven times every three days. (**A**) The expression of MMP-9 and CXCL-10 in tumor tissues and p-STAT5 in lymphocytes of tumor tissues assessed by Western blot. β-Actin was used as a loading control. Dark bands indicate protein expression. Quantitative analysis of the expression of (**B**) MMP-9, (**C**) CXCL10 and (**D**) p-STAT5 by Image pro plus (ChemiScope analysis, mean ± SEM, n = 3, * *p* < 0.05, ** *p* < 0.01, *** *p* < 0.001). (**E**) The proportions of CD3^+^ T-cells (CD3^+^), (**F**) CD4^+^ T-cells (CD3^+^CD4^+^), (**G**) CD8^+^ T-cells (CD3^+^CD8^+^) and (**H**) Treg cells (CD4^+^CD25^+^FoxP3^+^) in tumors were determined by flow cytometry (mean ± SEM, n = 3, * *p* < 0.05, ** *p* < 0.01).

**Figure 6 pharmaceutics-13-01990-f006:**
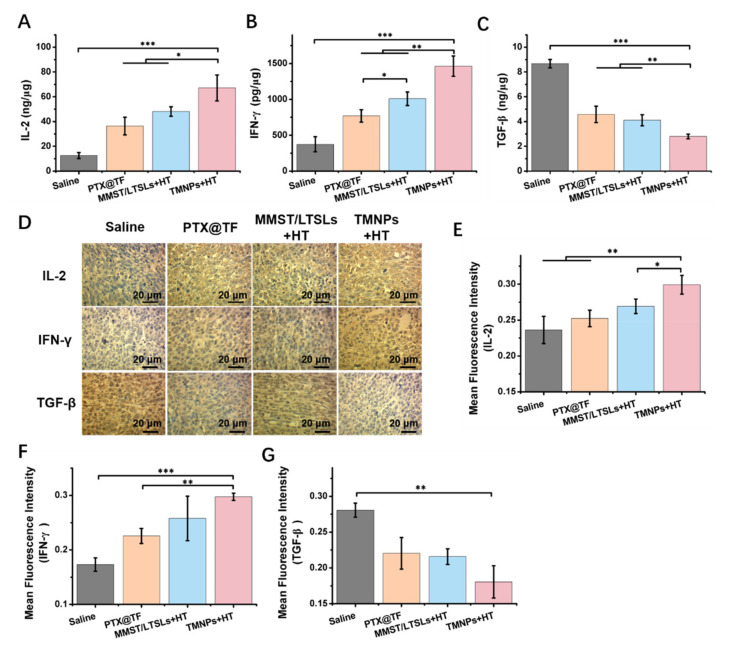
Alternation of immune-related cytokines by TMNPs in tumors. Quantitative analysis of (**A**) IL-2, (**B**) IFN-γ and (**C**) TGF-β in tumor tissues. Breast tumor fragments were harvested at day 29 after tumor implantation. Cytokines were measured by ELISA kit (mean ± SEM, n = 3, * *p* < 0.05, ** *p* < 0.01, *** *p* < 0.001). (**D**) Immunochemistry staining of IL-2, IFN-γ and TGF-β in tumor tissues. The scale bar is 20 μm. Integrated optical density (IOD) of (**E**) IL-2, (**F**) IFN-γ and (**G**) TGF-β in tumor tissues were quantified by Image Pro Plus (ImageJ 1.8.0, mean ± SEM, n = 3, * *p* < 0.05, ** *p* < 0.01).

**Figure 7 pharmaceutics-13-01990-f007:**
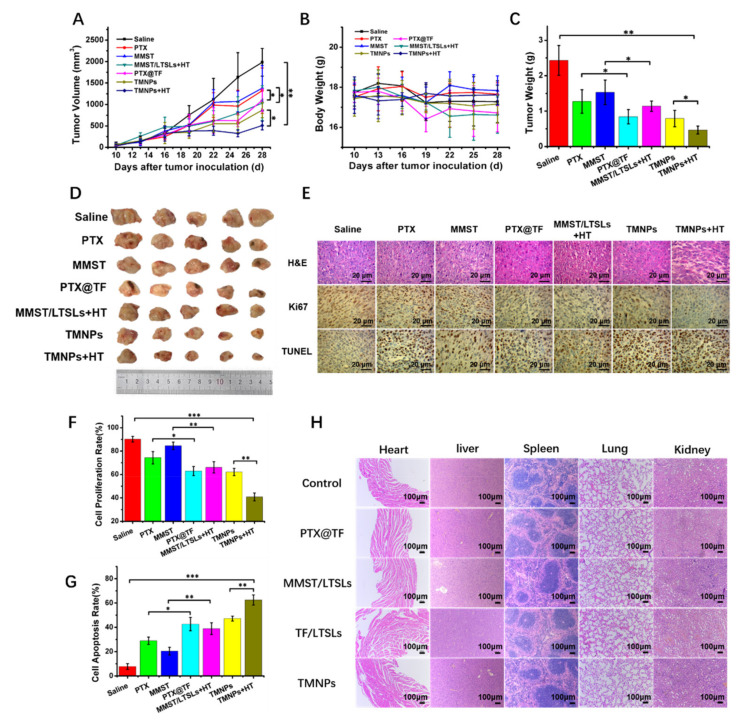
In vivo anti-tumor activity. The 4T1 tumor-bearing mice were treated with different samples (5 mg/kg PTX and/or 5 mg/kg MMST, n = 6) via tail vein injection at 10 days after tumor inoculation. The treatment was performed seven times every three days. (**A**) Tumor volume and (**B**) body weight growth curves after final treatment at day 29. (**C**) Tumor weight and (**D**) representative images of tumor tissues after treatment with different formulations at day 28 after tumor implantation (mean ± SEM, n = 6, * *p* < 0.05, ** *p* < 0.01). (**E**) Histological observation of the tumor tissues stained with H&E analysis, immunohistochemistry of Ki67 and TUNEL assay. The brown-staining cells represent the positive cells in the TUNEL and Ki67 assay. Nuclei were stained blue while the extracellular matrix and cytoplasm were stained red in the H&E analysis. The scale bar is 20 μm. Quantitative analysis of (**F**) proliferation and (**G**) cell apoptosis (mean ± SEM, *n* = 3, * *p* < 0.05, ** *p* < 0.01, *** *p* < 0.001). (**H**) Hematoxylin and eosin (H&E) stained organ slices from the normal BALB/c mice treated with PBS (control), MMST/LTSLs, PTX@TF, TF/LTSLs (blank carriers) and TMNPs (n = 5). The scale bar is 100 μm.

**Table 1 pharmaceutics-13-01990-t001:** Average particle size, PDI and surface potential of PTX@TF, MMST/LTSLs and TMNPs.

	PTX@TF	MMST/LTSLs	TMNPs
Size (nm) ± S.E.M	140.25 ± 2.5	89.07 ± 3.1	127.15 ± 3.5
PDI ± S.E.M	0.255 ± 0.02	0.211 ± 0.01	0.234 ± 0.08
Zeta potential (mV) ± S.E.M	−11.43 ± 1.10	4.01 ± 1.82	−1.62 ± 0.98

## Data Availability

The data presented in this study are available on request from the corresponding author. The data are not publicly available due to limited web resources.

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
