# Peer review of "Dual Targeting of Cancer Cells and MMPs with Self-Assembly Hybrid Nanoparticles for Combination Therapy in Combating Cancer"

_pharmaceutics, 2021, doi:10.3390/pharmaceutics13121990_

Round 1

Reviewer 1 Report

The authors created hybrid thermosensitive nanoparticles (TMNPs) composed of PTX@TF and MMST-loaded thermosensitive liposomes (MMST/LTSLs). They found that TMNPs promoted the intratumoral infiltration of CD4+ and CD8+T cells, reduced Treg, and suppressed tumor growth.

This is a novel drug delivery system that uses PTX, TF, MMST, and thermosensitive liposomes and may be useful in the development of breast cancer treatment.

Usually, TIMPs act to antagonize the activity of MMPs in the tumor microenvironment. Indeed, some researchers have reported that high TIMP-1 expression levels in breast cancer are associated with a poor prognosis for triple-negative breast cancer (Cheng-G Molecular Cancer 2016, 15:30). Expression of MMPs and TIMPs can be cell-dependent. Therefore, the effects of TMNPs should be tested on other breast cancer cell lines with different MMP and TIMP expression. It is also necessary to explain how baseline levels of TIMPs affect the antitumor effect of TMNPs.

The experimental conditions for thermotherapy in vivo are unknown. This needs to be explained in the method. How are the tissue temperature and the heated areas maintained optimally during the thermotherapy? How is the temperature monitored?

PTX@TF binds to TF receptor on tumor cells and is endocytosed, causing PTX to exert antitumor effects. The efficiency of PTX@TF uptake depends on the intensity of TF receptor expression in each cell. TF receptor is known to be highly expressed not only in proliferating cancer cells but also in hematogenic cells. The authors examined the accumulation of PTX@TF/DiR-LTSLs in the heart, liver, spleen, lungs, and kidneys. The effects on hematopoietic cells need to be examined.  TF receptors are highly expressed in growing cancer cells. If chemotherapy shrinks tumor cells and reduces TF receptor expression, TMNP's ability to bind and its antitumor effect will be diminished. In this situation, it is necessary to know the changes in TF receptor expression during treatment.

Figure 3G: The proportion of apoptotic cells was clearly increased in the LTSL groups with and without thermal treatment. The authors need to explain why.

Figure 4: The authors stated “large accumulation of the DiR was shown in normal tissues. It is mainly due to the characteristics of liposomes that make DiR-LTSls and PTX@TF/DiR-LTSLs phagocytosed by macrophage in some organs”. Do macrophages, activated lymphocytes and serum-induced fibroblasts that express TF receptor take up TMNPs and die by the action of PTX? The authors need to explain this possibility.

Figure 1: Rapture of TMNPs releases PTX, TF, and MMST into tumor microenvironment. Is the bond between PTX and TF preserved after thermal treatment? Otherwise, PTX cannot be selectively incorporated into tumor cells.

Differences in staining intensity in Figure 6D cannot be discerned at the cellular level. The method for quantifying the results of immunohistological staining (including related references) should be provided.

Legend in Figure 4: The authors stated “Tumor targeting ability of TMNPs in vivo”. However, TMNPs were not used in this experiment The authors need to reaffirm this point.

Page 2, line 87: It is necessary to explain why the formed hybrid nanoparticles can be abbreviated as TMNP.

Reviewer 2 Report

  • Add some quantitative results to the Abstract.
  • A better and updated literature review should be done in Introduction. I suggested the authors use the following ones:

DOIs: 10.1016/j.jconrel.2020.08.012 & 10.1063/1.5052455 & 10.1016/j.nantod.2020.101057 & …

  • Explain the idea behind the paper clearly at the last paragraph of the paper.
  • I didn’t follow “material and methods” section as there are 21 sub-sections. Please combine some of them with each other. Also, please add a flowchart describing step-by-step workflow of your paper.

Round 2

Reviewer 1 Report

The authors respond appropriately to the points raised by the reviewer.